# Multicentre double-blind randomised placebo-controlled trial evaluating the efficacy of the meningococcal B vaccine, 4CMenB (Bexsero), against *Neisseria gonorrhoeae* infection in men who have sex with men: the GoGoVax study protocol

Kate L Seib ,[1] Basil Donovan,[2] Caroline Thng,[1,3] David A Lewis,[4,5] Anna McNulty,[6,7] Christopher K Fairley,[8,9] Barbara Yeung,[2] Fengyi Jin,[2] Doug Fraser,[2] Benjamin R Bavinton,[2] Matthew Law,[2] Marcus Y Chen,[8,9] Eric P F Chow,[8,9] David M Whiley,[10] Brent Mackie,[11] Michael P Jennings,[1] Amy V Jennison ,[12] Monica M Lahra,[13,14] Andrew E Grulich [2]

For numbered affiliations see end of article.

**Correspondence to**
Dr Kate L Seib;
k.seib@griffith.edu.au

## ABSTRACT

**Introduction** Gonorrhoea, the sexually transmissible infection caused by *Neisseria gonorrhoeae*, has a substantial impact on sexual and reproductive health globally with an estimated 82 million new infections each year worldwide. *N. gonorrhoeae* antimicrobial resistance continues to escalate, and disease control is largely reliant on effective therapy as there is no proven effective gonococcal vaccine available. However, there is increasing evidence from observational cohort studies that the serogroup B meningococcal vaccine four-component meningitis B vaccine (4CMenB) (Bexsero), licensed to prevent invasive disease caused by *Neisseria meningitidis*, may provide cross-protection against the closely related bacterium *N. gonorrhoeae*. This study will evaluate the efficacy of 4CMenB against *N. gonorrhoeae* infection in men (cis and trans), transwomen and non-binary people who have sex with men (hereafter referred to as GBM+).

**Methods and analysis** This is a double-blind, randomised placebo-controlled trial in GBM+, either HIV-negative on pre-exposure prophylaxis against HIV or living with HIV (CD4 count >350 cells/mm³), who have had a diagnosis of gonorrhoea or infectious syphilis in the last 18 months (a key characteristic associated with a high risk of *N. gonorrhoeae* infection). Participants are randomised 1:1 to receive two doses of 4CMenB or placebo 3 months apart. Participants have 3-monthly visits over 24 months, which include testing for *N. gonorrhoeae* and other sexually transmissible infections, collection of demographics, sexual behaviour risks and antibiotic use, and collection of research samples for analysis of *N. gonorrhoeae*-specific systemic and mucosal immune responses. The primary outcome is the incidence of the first episode of *N. gonorrhoeae* infection, as determined by nucleic acid amplification tests, post month 4. Additional outcomes consider the incidence of symptomatic or asymptomatic *N. gonorrhoeae* infection at different anatomical sites (ie, urogenital, anorectum or oropharynx),

## STRENGTHS AND LIMITATIONS OF THIS STUDY

⇒ This is the first double-blind randomised placebo-controlled trial to evaluate the efficacy of the four-component meningitis B vaccine (4CMenB) against *Neisseria gonorrhoeae* infection in men who have sex with men and will provide information regarding vaccine efficacy against symptomatic and asymptomatic infection at different anatomic sites.

⇒ Antimicrobial resistance (AMR) testing and genome sequencing of *N. gonorrhoeae* positive samples will provide information regarding the circulating strains in the study population, and the vaccine efficacy against circulating strains including those with differing AMR profiles.

⇒ Immunological analysis of samples from the trial will provide insight into the *N. gonorrhoeae*-specific immune response following 4CMenB vaccination.

⇒ Limitations of the study are that it does not include cisgender women, immunosuppressed individuals living with HIV (CD4 count ≤350 cells/mm³) or participants with a low risk of *N. gonorrhoeae* infection.

incidence by *N. gonorrhoeae* genotype and antimicrobial resistance phenotype, and level and functional activity of *N. gonorrhoeae*-specific antibodies.

**Ethics and dissemination** Ethical approval was obtained from the St Vincent's Hospital Human Research Ethics Committee, St Vincent's Hospital Sydney, NSW, Australia (ref: 2020/ETH01084). Results will be disseminated in peer-reviewed journals and via presentation at national and international conferences.

**Trial registration number** NCT04415424.

## INTRODUCTION

Gonorrhoea is a global public health concern due to its prevalence, the severe sequelae it

can cause if left untreated, and the increasing difficulty in treating multidrug resistant strains of *Neisseria gonorrhoeae*.[1 2] There are estimated to be more than 82 million new cases of *N. gonorrhoeae* infections annually worldwide,[3] and the incidence is rising. Between 2012 and 2019, there was a 127% increase in notified gonorrhoea cases in Australia[4] and a 74% increase in the USA.[5] In Australia, the majority of gonococcal infections are in gay, bisexual and other men who have sex with men (GBM).[6] An Australian state-wide implementation project of pre-exposure prophylaxis (PrEP) against HIV infection in GBM documented a gonorrhoea incidence of 39.0 per 100-person years (PY), with infected sites including the urethra (7.3/100 PY), pharynx (19.7/100 PY) and anus (22.6/100 PY).[7] Analysis from sexual health clinics across Australia indicated that HIV-positive GBM had a gonorrhoea incidence of 29.1/100 PY.[8]

The outcomes of *N. gonorrhoeae* infection vary by site of infection and by sex.[1] Oropharyngeal and anorectal infections are typically asymptomatic. Symptomatic gonococcal infection typically presents as urethritis in males and cervicitis in females, although up to 10% of male and ~30%–80% of female genital tract infections are asymptomatic.[3] Undiagnosed or untreated urogenital tract infections can lead to severe sequelae, including urethral stricture, urogenital tract abscesses, epididymo-orchitis and infertility in males, and pelvic inflammatory disease, adverse pregnancy outcomes and infertility in females.[3] *N. gonorrhoeae* occasionally disseminates, resulting in sepsis, septic arthritis, perihepatitis, endocarditis and/or meningitis. Infection with *N. gonorrhoeae* may also increase the risk of acquiring and transmitting HIV.[1]

*N. gonorrhoeae* antimicrobial resistance (AMR) is of utmost concern, particularly with emerging resistance to ceftriaxone. In 2018, extensively drug-resistant (and ceftriaxone-resistant) *N. gonorrhoeae* were identified in Australia, as resistant to all commonly used antibiotics, and very recently further isolates were reported in the UK, Austria, Cambodia and France.[9–16] As such, *N. gonorrhoeae* has been prioritised as an urgent public health threat for which new preventive measures are needed, such as a vaccine.[17–19] Unfortunately, gonococcal vaccine development has been unsuccessful to date and there are large knowledge gaps regarding what constitutes a protective immune response for *N. gonorrhoeae*.[1] However, there is increasing evidence from observational cohort and case–control studies that outer membrane vesicle (OMV)-based *Neisseria meningitidis* (meningococcal) serogroup B (MenB) vaccines may provide cross-protection against gonorrhoea.[20–23] *N. gonorrhoeae* and *N. meningitidis* are closely related bacteria, with conservation of the majority of genes[24] and thus also conservation of outer membrane proteins that could be targeted by vaccines. Observational studies have detected a decrease in the incidence of *N. gonorrhoeae* infection following vaccination with the OMV meningococcal vaccines MeNZB[20] or four-component meningitis B vaccine (4CMenB)[21–23] with vaccine effectiveness estimated to be approximately 30%–50%. The

OMVs in these vaccines are structures that are released from the outer membrane of Gram-negative bacteria and contain phospholipids, lipooligosaccharide and a complex mix of proteins. The broader spectrum four-component serogroup B vaccine, 4CMenB (marketed as Bexsero) contains the MeNZB OMV antigens plus an additional three recombinant antigens (NHBA-GNA1030 and fHbp-GNA2091 fusions and NadA).[25] There are homologues of 20 of the 22 major OMV vaccine proteins present in *N gonorrhoeae*, as well as the NHBA antigen.[26] Therefore, several vaccine antigens may be responsible for inducing an immune response that cross-reacts with *N. gonorrhoeae*. The observational study findings with MenB OMV-based vaccines are the first time that any vaccine has been associated with protection against gonorrhoea. However, data from a randomised control trial are required to confirm the efficacy of 4CMenB to protect against *N. gonorrhoeae* infection.

### Study aims and objectives

This study aims to evaluate the efficacy of 4CMenB in the prevention of *N. gonorrhoeae* infection in men (cis and trans), transwomen, transmen and non-binary people who have sex with men (hereafter referred to as GBM+). The primary outcome is the incidence of the first episode of *N. gonorrhoeae* infection, as determined by nucleic acid amplification tests (NAATs), post month 4 (ie, after 4 weeks post dose 2). Secondary outcomes are the incidence of symptomatic or asymptomatic *N. gonorrhoeae* infection at different anatomical sites (ie, urogenital, anorectum or oropharynx), incidence by *N. gonorrhoeae* genotype and AMR phenotype, and level and functional activity of any induced *N. gonorrhoeae*-specific antibodies. Primary, secondary and immunogenicity objectives and endpoint measures are outlined in table 1.

### METHODS AND ANALYSIS
### Study design and setting

The study is a double-blind, randomised placebo-controlled trial of a two-dose schedule of 4CMenB to prevent urethral, oropharyngeal and anal *N. gonorrhoeae* infection in GBM+, who are either taking HIV PrEP or postexposure prophylaxis (PEP), are planning to start PrEP after the cessation of PEP, or are living with HIV with a CD4 count >350 cells/mm³. Participants are randomised 1:1 to receive two doses of 4CMenB or saline placebo 3 months apart. Participants are required to attend 3-monthly follow-up visits over 24 months. At each visit, there is testing for *N. gonorrhoeae* and other sexually transmissible infections (STIs) including HIV, collection of blood and oral mucosal exudate samples for analysis of *N. gonorrhoeae*-specific immune responses, and collection of demographics, sexual behaviour risks and antibiotic use. An overview of the study design is shown in figure 1 and a summary of the study visits is shown in table 2.

The study is being conducted at five public sexual health clinics and two private clinics in New South Wales,

**Table 1**  Primary, secondary and immunogenicity objectives and outcome measures of the GoGoVax study

| Objective* | Outcome measures |
|---|---|
| **Primary** | |
| To investigate whether the 4CMenB vaccine, when administered in a two-dose regimen at 0 and 3 months, reduces the incidence of the first episode of *Neisseria gonorrhoeae* infection, as determined by NAAT testing of urine samples, or swabs taken from the urethra, anorectum, oropharynx and vagina, in GBM+populations regardless of whether the infection is symptomatic or asymptomatic. | Detection of *N. gonorrhoeae* infection in the first instance in a urine sample or on a swab taken from the urethra, anorectum, oropharynx or vagina, as determined by NAAT testing, post month 4. |
| **Secondary** | |
| To investigate the impact of administration of a two-dose regimen of 4CMenB vaccine on the incidence of the first episode of symptomatic *N. gonorrhoeae* infection of the urethra, anorectum or vagina, as determined by NAAT testing. | Detection of *N. gonorrhoeae* infection in the first instance in the urethra, anorectum or vagina, as determined by NAAT testing, post month 4, at a study visit when a participant also reports any symptoms at the relevant anatomic site. |
| To investigate the impact of a two-dose regimen of 4CMenB vaccine on the incidence of the first episode of asymptomatic *N. gonorrhoeae* infection of the urethra, anorectum, oropharynx or vagina, as determined by NAAT testing. | Detection of *N. gonorrhoeae* infection in the first instance in the urethra, anorectum, oropharynx or vagina, as determined by NAAT testing, post month 4, at a study visit when a participant reports no symptoms at the relevant anatomic site. |
| To investigate the impact of administration of a two-dose regimen of 4CMenB vaccine on the incidence of first episode of *N. gonorrhoeae* infection, as determined by NAAT testing, regardless of symptoms and anatomic sites, by various *N. gonorrhoeae* strain types (genotype and AMR phenotype). | Detection of *N. gonorrhoeae* infection in the first instance in the urethra, anorectum, oropharynx, or vagina, as determined by NAAT testing post month 4, by genotype and AMR phenotypes. |
| **Immunogenicity** | |
| To evaluate if the *N. gonorrhoeae*-specific ELISA, serum bactericidal activity assay (SBA) and/or opsonophagocytic killing assay (OPK) titres increase following 4CMenB vaccination. | The ELISA, SBA and OPK titres of serum and ELISA titres of oral mucosal transudates post 4CMenB dose 2, relative to baseline. |
| To evaluate if the *N. gonorrhoeae*-specific ELISA, SBA and/or OPK titres correlate with reduced *N. gonorrhoeae* infection. | The ELISA, SBA and OPK titres of serum during the study period. |

*For all primary and secondary efficacy objectives, additional analyses will be conducted to allow repeated diagnoses of *N. gonorrhoeae* infection, as determined by NAAT testing from the same individuals to be included. This is to compare the overall incidence of all episodes of *N. gonorrhoeae* infection diagnosed during the study period between the vaccine and placebo arms, allowing multiple diagnoses of *N. gonorrhoeae* infection occurring in the same individuals at different time points.
AMR, antimicrobial resistance; 4CMenB, four-component meningitis B vaccine; NAAT, nucleic acid amplification test.

Victoria and Queensland, Australia: the Sydney Sexual Health Centre, the Western Sydney Sexual Health Centre, the Melbourne Sexual Health Centre, the Gold Coast Sexual Health Service, Royal Prince Alfred Sexual Health Sydney, Taylor Square Private Clinic Sydney and Prahran Market Clinic Melbourne.

### Study population, recruitment and eligibility criteria

The participant population for this trial is GBM+taking PrEP/PEP or living with HIV (CD4 count >350 cells/mm³), who have had a diagnosis of gonorrhoea or infectious syphilis in the last 18 months (a key characteristic associated with greater gonorrhoea incidence[7]). This population is recommended under Australian guidelines to attend clinics 3 monthly for comprehensive sexual health screening, including NAAT-based screening of urogenital, oropharyngeal and anal specimens for *N. gonorrhoeae* infection.[6 27]

Potentially eligible participants were identified by clinicians at each clinic based on the inclusion and exclusion criteria (see box 1) and were referred to the study clinicians or study nurses who explained the study, checked eligibility and obtained informed consent. Potentially eligible participants were also referred to the participating clinics by other public sexual health centres, public health clinics, general practices, and state-based lesbian, gay, bisexual, and transgender community organisations, such as ACON and Positive Life New South Wales (NSW) in NSW, Thorne Harbour Health and Living Positive Victoria in Victoria, and the Queensland Council for LGBTI Health and Queensland Positive People.

### Study processes
#### Randomisation and blinding
Once recruited, participants were randomly assigned (1:1) and stratified by clinical sites via a web-based

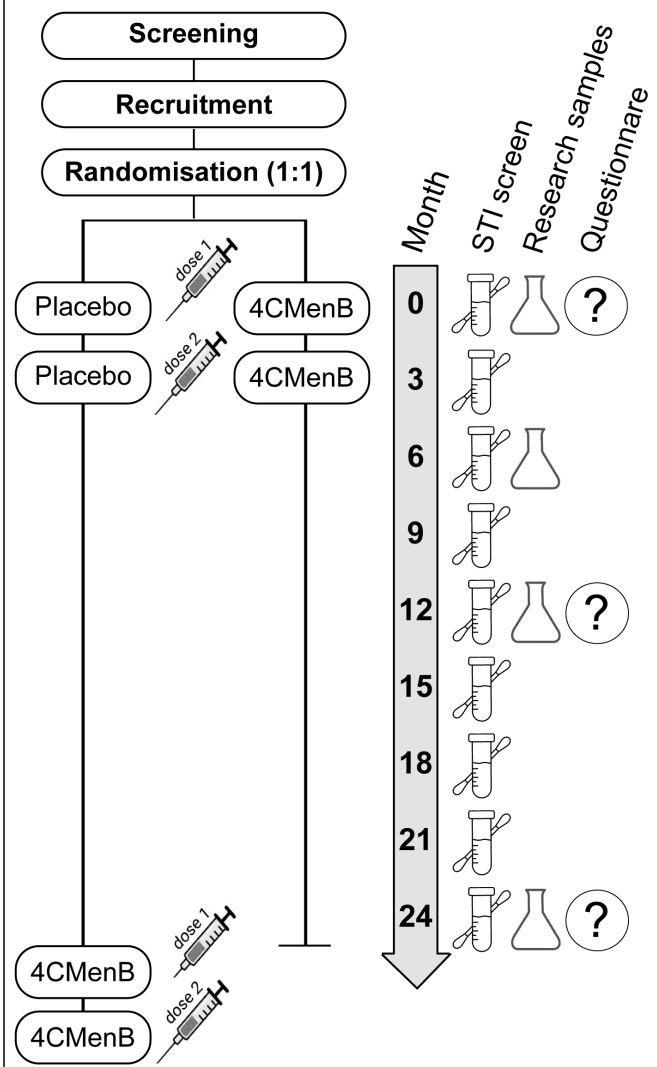

**Figure 1** Study design of the GoGoVax trial. 4CMenB, four-component meningitis B vaccine; STI, sexually transmissible infection.

randomisation system, to receive either 4CMenB or saline placebo. Participants, their clinicians (site investigators/doctors, site coordinators/nurses) and study researchers assessing the outcomes were blinded to the assignment. The site pharmacists and drug manager who dispensed the study treatment, and the vaccinators who administered the study treatment were unblinded to the treatment allocation.

### Intervention

A prevaccination checklist was completed prior to administration of the intervention to ensure the safety of participants. At the month 0 (baseline) and month 3 visits participants received either a dose of 0.5 mL 4CMenB or a dose of 0.5 mL placebo (150 mmol of sodium chloride/0.9% saline solution) by intramuscular injection. This vaccine schedule is consistent with 4CMenB's indication for immunisation against invasive disease caused by *N. meningitidis* group B in people >2 years of age (two doses with an interval of ≥1 month).[28] 4CMenB is a

licensed vaccine in Australia with well-documented safety profile but it is not approved by the Therapeutic Good Administration (TGA) for the prevention of gonorrhoea. Therefore, the trial is being conducted under the Clinical Trial Notification scheme. At the completion of the study, participants who were randomised to the placebo arm will be offered the 4CMenB vaccine.

### Sample and data collection

The study assessments, samples and data collected at each 3-monthly visits over 24 months are outlined in figure 1 and table 2. At each 3-monthly visit, study participants have routine specimens collected as part of their PrEP or HIV clinical care as per the Australian STI Management Guidelines.[29] These specimens include urogenital, oropharyngeal and anorectal specimens for screening for *N. gonorrhoeae* and *Chlamydia trachomatis* by NAAT, and blood for screening for HIV and syphilis. At each visit, data are also recorded regarding symptoms that may be related to gonorrhoea including urethritis, proctitis, epididymitis and cervicitis/vaginitis. Unscheduled visits to the participating sites or external healthcare providers, at which gonorrhoea is diagnosed by an NAAT, are also recorded.

Participants complete a study questionnaire at baseline to collect demographics and sexual risk behaviour. A questionnaire is also completed at months 12 and 24 to collect ongoing sexual risk behaviour. At each visit, recent antibiotic use and history of meningococcal B vaccination within the last 3 months are also recorded.

Research specimens (blood and oral mucosal transudates) are collected at baseline, and at months 0, 6, 12 and 24 at selected study sites to investigate vaccine-induced antibodies via assays including ELISA, serum bactericidal activity, opsonophagocytic killing and antigen function blocking assays.[26 30 31] Culture isolates collected as part of the routine care for participants with *N. gonorrhoeae* infection will be used to conduct phenotypic AMR testing[32 33] and whole genome sequence analysis.[34 35] Testing will be performed at the WHO Collaborating Centre for STIs and AMR at the Prince of Wales Hospital in Sydney, the Queensland Forensic and Scientific Services at Queensland Health in Brisbane, and/or at the Microbiological Diagnostic Unit in Melbourne.

### Safety reporting

4CMenB is a licensed vaccine with a well-documented safety and tolerability profile.[36] Therefore, safety monitoring is focused on serious adverse events (SAEs). All SAEs will be reported within 24 hours to the study sponsor and the project medical officer (at the Kirby Institute) and to the Human Research Ethics Committee (HREC) annually. The sponsor will notify the TGA, HREC and site principal investigators of all significant events that can adversely affect the safety of participants.

**Table 2** Summary of study visits

| Study visit | Screening* | Baseline (month 0) | Month 3 (±14 days) | Month 6 (±14 days) | Month 9 (±14 days) | Month 12, 15, 18, 21 (±14 days) | Final visit month 24 (±14 days) | Unscheduled† |
|---|---|---|---|---|---|---|---|---|
| **Study assessments** | | | | | | | | |
| Informed consent | X | | | | | | | |
| Review eligibility | X | | X | | | | | |
| Risk assessment—gonorrhoea or syphilis in the last 18 months | X | | | | | | | |
| Relevant medical history‡, demographics | X | | | | | | | |
| Concomitant antibiotic use, meningococcal B vaccination | X | | X | X | X | X | X | X |
| Symptoms of urethritis, proctitis, epididymitis and cervicitis/vaginitis | X | | X | X | X | X | X | X |
| Prevaccination checklist | | X | X | | | | | |
| Vaccine or placebo administration | | X | X | | | | | |
| Serious adverse events review | | X | X | X | X | | | |
| Study questionnaire | | X | | | | X§ | X | |
| Randomisation¶ | | X | | | | | | |
| **Routine laboratory assessments** | | | | | | | | |
| HIV antibody test** | X | | X | X | X | X | X | |
| HIV viral loads, CD4 count†† | X | | | | | | | |
| STI tests‡‡ – urine, swabs and blood collection | X§§ | | X | X | X | X | X | X |
| Pregnancy test¶¶ | X | X | | | | | | |
| **Research specimens for central laboratory** | | | | | | | | |
| Blood specimen (at selected study sites)—two 8 mL samples (for Neisseria-specific immune response) | | X*** | | X | | X††† | X | |
| Oral mucosal transudates (at selected study sites, for Neisseria-specific immune response) | | X*** | | X | | X††† | X | |
| Culture isolates and NAAT samples (gonorrhoea infection incident cases only) | X | | X | X | X | X | X | X |

*All screening assessments are to be completed within 14 days of baseline (the visit when the first dose of 4CMenB or matched placebo will be administered). The reason participants failed screening or were ineligible to be randomised or take part in the study is recorded.
†Unscheduled visit—an unscheduled visit will be conducted for (1) when a participant is diagnosed with symptomatic gonorrhoea infection by an NAAT test; (2) when a participant is diagnosed with asymptomatic gonorrhoea infection by an NAAT test or (3) when a participant returns for test of cure and has a positive gonorrhoea NAAT test.
‡Including vaccination history for 4CMenB vaccine.
§Study questionnaire—month 12 only.
¶Randomisation can occur anytime between screening and baseline visit when the required HIV test result(s) and drug kit (containing the 4CMenB vaccine or matched placebo) are available, and an individual is confirmed to be eligible for the trial. Randomisation can occur on the same day of screening if the above aforementioned criteria are met.
**HIV negative participants only. An HIV negative antibody test within 4 months of screening can be used for randomisation if the participant has been taking PrEP (daily PrEP or on-demand PrEP) consistently. An individual starting PEP who has tested HIV negative within 4 months can be enrolled without waiting for the HIV antibody test result taken at PEP initiation. In the rare occurrence that the HIV tests taken that day returns positive, and the participant has already had their first vaccine or placebo dose, the participant should not receive the second vaccine or placebo dose until their HIV viral load is <200 copies/mL and their CD4 is >350 cells/mm³.
††HIV-positive participants only. An undetectable HIV viral load of <200 copies/ml and CD4 count of >350 cells/mm³ can be used for randomisation if the tests were conducted within 12 months of screening.
‡‡Three-monthly STI tests as per the Australasian Society for HIV, Viral Hepatitis and Sexual Health Medicine (ASHM), Australian STI Management Guidelines For Use In Primary Care, *ASHM*, http://www.sti.guidelines.org.au/.
§§STI tests conducted within 14 days of screening do not have to repeated and the results can be used for screening.
¶¶Pregnancy test for trans men and non-binary people who were recorded female at birth.
***Research specimens at baseline to be taken before the vaccine or placebo is given.
†††Month 12 only.
4CMenB, four-component meningitis B vaccine; NAAT, nucleic acid amplification test; PEP, postexposure prophylaxis; PrEP, pre-exposure prophylaxis; STI, sexually transmissible infection.

## Data management

Source documents include but are not limited to participant medical records, laboratory reports, participant progress notes, pharmacy records and any other reports or records of procedures performed in accordance with the protocol. Study data are collected using the Medrio Electronic Data Capture system (Medrio, San Francisco, California, USA). Medrio is a cloud-based, web-enabled password-protected platform. Each participant is assigned a unique participant identification number that is documented in the participant's medical record and all study documents including the dispensing records. Data are stored using Google Cloud Platform facilities globally. Medrio is adherent to ICH GCP, FDA 21 CFR Part 11; EudraLex, Annex 11; GDPR; and HIPAA. All site data entry personnel will be required to pass training and assessments to be able to enter data and all will have their confidential login credentials. The login password will be changed every 3 months. Following each participant visit, the designated site staff complete the visit-specific

## Box 1 Study inclusion and exclusion criteria

Inclusion criteria

1. Between 18 and ≤50 years of age.
2. Men (cis and trans), transwomen and non-binary people who have had sex with at least one man in the last 6 months.
3. Diagnosis of gonorrhoea or infectious syphilis in the last 18 months.
4. Committed not to take doxycycline as prophylaxis for the duration of the trial.*
5. Able to understand spoken and written English.
6. Willing and likely to comply with the trial procedures for 2 years.
7. Agree to be contacted via SMS/phone/email by the study team and either
   a. HIV-negative (with an HIV negative antibody test within 4 months of screening) and taking HIV pre-exposure prophylaxis (PrEP) (daily PrEP or on-demand PrEP) within the last 4 months at the time of enrolment or taking HIV postexposure prophylaxis (PEP) and there is a plan to start PrEP after the cessation of PEP.
   b. HIV-positive and on an antiviral regimen, with an undetectable virus level of <200 copies/mL and a CD4 count >350 cells/mm$^3$ (to optimise the immune response to vaccine) within 12 months of screening.

Exclusion criteria

1. Have a history of vaccination for meningococcal B with four-component meningitis B vaccine.
2. Have contraindications to receiving the meningococcal B vaccine, which include:
   – Anaphylaxis following a previous dose of any meningococcal vaccine.
   – Anaphylaxis following any vaccine component.
3. Are participating in biomedical prevention strategies for bacterial sexually transmissible infections (participation in diagnostic or treatment studies is not an exclusion).
4. Are taking long-term (>4 weeks) antibiotic for prophylaxis or treatment for acne, malaria, syphilis or other bacterial condition(s).
5. Have defects in, or deficiency of, complement components, including factor H, factor D or properdin deficiency.
6. Are taking or will receive complement inhibitors such as eculizumab (a monoclonal antibody directed against complement component C5) or ravulizumab.
7. Have functional or anatomical asplenia, including sickle cell disease or other haemoglobinopathies, and congenital or acquired asplenia.
8. Have had a haematopoietic stem cell transplant.
9. Have any major unstable medical condition or therapy that may cause immune compromise (eg, chemotherapy, radiation, corticosteroids (prednisone >5 mg/day) within 14 days prior to screening).
10. Documented allergy to latex and/or kanamycin.
11. Have prior known meningococcal disease.
12. Positive pregnancy test at screening.

*In accordance with the Australasian Society for HIV, Viral Hepatitis and Sexual Health Medicine (ASHM) 2023 Consensus Statement on doxycycline prophylaxis (Doxy-PEP) (https://ashm.org.au/about/news/doxy-pep-statement/), site clinicians were advised that they could prescribe doxycycline to study participants if they felt this was appropriate for their patients, and the number of days participants have taken doxycycline is recorded at each visit.

electronic case report form as soon as possible after the completion of the visit. There will be no personal information or fully identifiable information of any participants entered or stored in the study database. Participants also provide consent for the project team to acquire any other information and results from other health services for the purpose of the research via linkage with the Australian Collaboration for Coordinated Enhanced Sentinel Surveillance (ACCESS) system.[37] Data from ACCESS will enable long-term follow-up after the completion of the study even if a participant moves to another clinic. At The Kirby Institute, UNSW Sydney (sponsor), designated research personnel have viewing access to the EDC to monitor data and conduct source data verification. The study data will be extracted after the database is closed at the end of the study. The study data will be kept for 15 years after study completion.

### Sample size

The primary endpoint of the study will be the incidence of the first episode of *N. gonorrhoeae* infection (urogenital, anal or oropharyngeal) as determined by NAAT testing, per 100 PY, post month 4. Assuming a vaccine efficacy (VE) of 30% (HR=0.7; as predicted for MeNZB),[20] and a conservative *N. gonorrhoeae* incidence of 25 per 100 PY (based on recent Australian data for symptomatic and asymptomatic infections in GBM on PrEP (39 per 100 PY)[7] and in HIV-positive GBM (29 per 100 PY)[8]), 1:1 randomisation, a 12-month recruitment period, and 24-month follow-up, and 10% lost to follow-up, a total of 730 participants (365 randomised to each arm) would need to be recruited to have 80% power to demonstrate VE (two-sided alpha=5%).

### Statistical analysis

The primary efficacy endpoint is the incidence of the first episode of *N. gonorrhoeae* infection detected by an NAAT, post month 4. A secondary endpoint will include the incidence of all episodes of *N. gonorrhoeae* infection detected by an NAAT, including those within 4 weeks post vaccine dose 2. All episodes of chlamydial infection diagnosed during the study period will also be captured for the proposed additional analyses to compare the incidence of chlamydial infection between the vaccine arm and the placebo arm.

### Analysis populations

The primary analysis of efficacy will be per-protocol, including study participants who received two vaccine or placebo doses, have two or more follow-up visits following the second dose and do not fail on any exclusion criterion pertinent to the assessment of efficacy. In this analysis, only *N. gonorrhoeae* infections 4 weeks postdose 2 (ie, after month 4) will be considered. A modified intention-to-treat analysis will be performed as a secondary analysis including all participants (including general protocol violators) who receive at least one dose of vaccine or placebo and have at least 6 months of study follow-up after the first dose. For these analyses, any efficacy against cases occurring after the day 1 visit will be counted.

## Statistical methods

VE is expressed as a percentage and will be defined as:

$$VE = 100\{1 - (r_v/r_p)\}$$

where $r_v$ is the incidence rate among vaccine recipients and $r_p$ is the incidence rate among placebo recipients. Each efficacy hypothesis will be tested by constructing a two-sided 95% CI for VE using Poisson regression models, accommodating repeated measures for the secondary endpoint.

## Patient and public involvement

Representatives from the community groups ACON (an LGBTQ+ health organisation) and Positive Life NSW (a community organisation for people living with HIV) have been involved in the research since the initial funding application and have participated in monthly/2-monthly management team meetings since the start of trial planning. They provided input on the protocol, study information and recruitment materials. These community partners will assist in the dissemination of study results to participants and the community via their social media platforms.

## DISCUSSION

In light of increasing gonorrhoea incidence and escalating AMR, there is an urgent need for new gonorrhoea prevention strategies. The WHO has set a target for reducing gonorrhoea incidence by 90% by 2030 and highlighted the need for an effective vaccine to help achieve this goal.[38] GoGoVax is the first double-blind, randomised placebo-controlled trial to investigate the efficacy of the meningococcal 4CMenB vaccine against *N. gonorrhoeae* infection in GBM+. This trial will provide critical scientific evidence to inform potential indication expansion of 4CMenB to include gonorrhoea, which could lead to a substantial reduction in *N. gonorrhoeae* incidence worldwide. Observational evidence to date suggests that 4CMenB may have vaccine effectiveness against *N. gonorrhoeae* of approximately 30%–50%.[21–23] Mathematical modelling in an MSM population indicates that a gonococcal vaccine with 25%–50% efficacy and 2-year duration of protection could reduce gonorrhoea prevalence by 31%–62% in 2 years if 30% of MSM presenting for STI screening are vaccinated per visit.[39]

Our study addresses several questions in the field regarding overall protection against *N. gonorrhoeae* infection provided by 4CMenB, but also regarding VE against symptomatic and asymptomatic infection at genital and extragenital sites of infection, efficacy against diverse strains, as well as duration and potential mechanisms of vaccine-mediated protection. The link between symptoms and *N. gonorrhoeae* load and transmission is unclear[40–42] and it is unknown if VE will differ depending on the presence or absence of symptoms. VE may also differ by anatomical site of infection. This has been seen for antibiotic treatment of gonorrhoea, with antibiotics being less effective against oropharyngeal gonorrhoea than for urethral and rectal infections.[43 44] Although 4CMenB can prevent invasive meningococcal disease, it is not believed to result in a clinically significant reduction in nasopharyngeal carriage of *N. meningitidis*.[45–47] However, unlike *N. meningitidis*, *N. gonorrhoeae* does not have a polysaccharide capsule, which may alter its susceptibility to 4CMenB vaccine-induced immune responses in the pharynx. Given the diversity of *N. gonorrhoeae*, the AMR testing and genome sequencing of *N. gonorrhoeae* positive samples will provide information regarding the circulating strains in the population to enable evaluation of VE against circulating strains, including differing AMR profiles. The linkage between our study and the Australian ACCESS surveillance system[37] will allow assessment of the durability of any preventive effect of 4CMenB beyond the duration of the trial. Immunological analysis of samples from the trial will provide insight into the *N. gonorrhoeae*-specific immune response following 4CMenB vaccination. Immunological correlates of protection for *N. gonorrhoeae* have not yet been identified,[48 49] and insight from this trial will facilitate future vaccine development and evaluation.

This trial is targeted at the people at the highest risk. Men who have sex with men and people living with HIV are key populations at higher risk for gonococcal infection[48] and Australian GBM+ are diagnosed with some of the highest gonorrhoea incidence rates ever described.[7 8] As such, it is essential to understand whether 4CMenB can prevent or reduce *N. gonorrhoeae* in this population. However, it should be noted that we did not enrol immunosuppressed men living with HIV (CD4 count <350 cells/mm$^3$) in order to optimise the immune response to vaccine in participants. Another important limitation of our study is the lack of inclusion of cisgender women. Gonorrhoea causes a significant burden of disease in females, and it is possible that past *N. gonorrhoeae* infection may impact vaccine-induced immune responses. Additional 4CMenB efficacy trials currently underway in different settings that include female participants (eg, ClinicalTrials.gov NCT04350138) will hopefully address some of these limitations and complement findings from our study.

Due for completion in 2025, the GoGoVax study will inform future gonorrhoea prevention strategies. If 4CMenB is proven to be effective against *N. gonorrhoeae*, this could facilitate changes in gonorrhoea prevention strategies that could improve sexual and reproductive health worldwide.

## Ethics and dissemination

The current study protocol (version 5, 24 June 2022) informed consent forms, and all recruitment materials have been reviewed and approved by St Vincent's Hospital Human Research Ethics Committee, St Vincent's Hospital Sydney, NSW, Australia (ref: 2020/ETH01084). The study results will be published in international peer-reviewed journals, presented at national and international conferences, and provided to study participants and the public via social, television, radio and/or print media.

**Author affiliations**
[1]Institute for Glycomics, Griffith University, Gold Coast, Queensland, Australia
[2]The Kirby Institute, University of New South Wales, Sydney, New South Wales, Australia
[3]Gold Coast Sexual Health, Gold Coast Hospital and Health Service, Southport, Queensland, Australia
[4]Western Sydney Sexual Health Centre, Sydney, New South Wales, Australia
[5]Sydney Medical School - Westmead, Faculty of Medicine and Health and Sydney Infectious Diseases Institute, University of Sydney, Sydney, New South Wales, Australia
[6]Sydney Sexual Health Centre, Sydney, New South Wales, Australia
[7]School of Population Health, University of New South Wales, Sydney, New South Wales, Australia
[8]Melbourne Sexual Health Centre, Alfred Health, Melbourne, Victoria, Australia
[9]School of Translational Medicine, Faculty of Medicine, Nursing and Health Sciences, Monash University, Melbourne, Victoria, Australia
[10]UQ Centre for Clinical Research, Faculty of Medicine, The University of Queensland, Brisbane, Queensland, Australia
[11]ACON Health, Sydney, New South Wales, Australia
[12]Public Health Microbiology, Queensland Health Forensic and Scientific Services, Brisbane, Queensland, Australia
[13]WHO Collaborating Centre for STI and AMR, New South Wales Health Pathology Microbiology, The Prince of Wales Hospital, Sydney, New South Wales, Australia
[14]UNSW Medicine, The University of New South Wales, Sydney, New South Wales, Australia

**Acknowledgements** We would like to acknowledge the GoGoVax Protocol Steering Committee members and the site investigators and coordinators. Protocol steering committee members: Professor KLS, Professor BD, Scientia Professor AG, Professor CKF, Professor DAL, Assistant Professor AM, Dr CT, Assistant Professor MP, Assistant Professor DW, Professor ML, Assistant Professor EC, Dr RF, Dr MR, Dr NC, Dr RV, Dr NR, Dr BE, Prof DT, Dr AVJ, Mr BM, Mr AH, Ms Jane Costello, Dr Christine Selvey, Prof Deborah Williamson, Professor John Kaldor, Assistant Professor Marcus Chen, Professor Michael Jennings, Professor Monica Lahra, Professor Rebecca Guy, Dr Fengyi Jin, Dr Benjamin Bavinton, Ms Barbara Yeung, Mr Doug Fraser, Ms Michelle Burns. Site investigators and coordinators: Sydney Sexual Health Centre—Assistant Professor Anna McNulty, Dr Rick Varma, Mr Paul Robinson, Mr Anik Ray, Ms Alison Jenkins. Western Sydney Sexual Health Centre—Assistant Professor David Lewis, Dr Nick Comninos, Dr Rohan Bopage, Ms Melissa Power, Mr Pradeep Kumar. Melbourne Sexual Health Centre—Professor Christopher Fairley, Assistant Professor Eric Chow, Assistant Professor Marcus Chen, Ms Kate Maddaford, Mr Finn Mercury. Gold Coast Sexual Health Services - Dr Caroline Thng, Ms Catherine Donald, Helen McEvoy. RPA Sexual Health—Professor David Templeton, Dr David Atefi, Dr Rachel Burdon, Dr Lara Whitbourne, Ms Linda Garton, Ms Ashlee Jackson. Taylor Square Private Clinic - Dr Rob Finlayson, Dr Mark O'Reilly, Ms Piper Calleia, Mr Kiran Muthukrishnan. Prahran Market Clinic—Dr Norman Roth, Dr Beng Eu, Ms Helen Lau.

**Contributors** KLS, BD and AEG designed the study and wrote the initial manuscript. KLS, BD, CT, DAL, AM, CKF, BY, FJ, DF, BRB, ML, MYC, EPFC, DMW, BM, MPJ, AVJ, MML and AEG contributed to study design, protocol development and manuscript review and approved the final version of the manuscript for submission.

**Funding** This work was supported by the Australian National and Medical Health Council (NHMRC) grant number 1182443. Additional support was provided by Griffith University (Administrative Institute) and the Kirby Institute UNSW Sydney (Study Sponsor). Vaccine was provided by GlaxoSmithKline (GSK) Australia. KLS acknowledges supported from the NHMRC (grants 2017383 and 2002182). EPFC is supported by an NHMRC Emerging Leadership Investigator Grant (GNT1172873).

**Disclaimer** The funding bodies, study sponsor and GSK had no role in the design of the study.

**Competing interests** None declared.

**Patient and public involvement** Patients and/or the public were involved in the design, or conduct, or reporting, or dissemination plans of this research. Refer to the Methods section for further details.

**Patient consent for publication** Not applicable.

**Provenance and peer review** Not commissioned; externally peer reviewed.

**ORCID iDs**
Kate L Seib http://orcid.org/0000-0002-7094-3528
Amy V Jennison http://orcid.org/0000-0002-5599-7480
Andrew E Grulich http://orcid.org/0000-0002-3269-1032

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
