## [Reviewer comments · BMJ Open]

ARTICLE DETAILS

TITLE (PROVISIONAL)	A multicentre double-blind randomised placebo-controlled trial evaluating the efficacy of the meningococcal B vaccine, 4CMenB (Bexsero), against Neisseria gonorrhoeae infection in men who have sex with men: the GoGoVax study Protocol
AUTHORS	Seib , KL; Donovan, Basil; Thng, Caroline; Lewis, David; McNulty, Anna; Fairley, Christopher; Yeung, Barbara; Jin, Fengyi; Fraser, Doug; Bavinton, BR; Law, Matthew; Chen, Marcus; Chow, Eric; Whiley, David; Mackie, Brent; Jennings, Michael; Jennison, Amy; Lahra, Monica M; Grulich, Andrew

VERSION 1 – REVIEW

REVIEWER	van Dam, Alje Amsterdam UMC Locatie AMC
REVIEW RETURNED	02-Jan-2024

GENERAL COMMENTS	The importance of the study is clear, and the protocol is appropriate. I have three comments: 1. Regarding inclusion criterium 4, committed not to take doxycycline as prophylaxis for the duration of the trial - this is a problem with regard to the 2023 Consensus Statement on doxycycline prophylaxis (Doxy-PEP) for the prevention of syphilis, chlamydia and gonorrhoea among gay, bisexual, and other men who have sex with men in Australia. Many participants may be eligible for doxycylin prophylaxis.2. Vaccine effectivity can be calculated from this trial. However vaccine effectivity is expected to be only 30-50%. To study the effectivity of the vaccine for public heath purposes it is important to know who is willing to take the vaccine. In the present study, no attempt is made to identify specific details of the population taking part in the study in comparison to those who were asked to take part but declined to participate. It could be useful to compare those groups by handing over a study questionnaire to persons eligible for the trial, but not inclined to take part.3. NAAT specimens could be stored to sequence certain genes of interest of strains that could not be cultured. I suppose this will be done by the consortium but this could be stated if so.
--

REVIEWER	Paynter, Janine University of Auckland, School of Population Health
REVIEW RETURNED	05-Jan-2024

GENERAL COMMENTS	This is an important well-designed study. Can you add more detail about data on participants who get a confirmed infection within the 4-week post dose 2 period please? Are they excluded or included?
---

	Vaccine efficacy equation should the percent sign be there? I don't think it should - the multiplication should simply be by 100 i.e. $100(1-(Rv/Rp))$ you could then note that VE is expressed as a percent. It's the same as adding any other unit e.g. ml to an equation. Maybe a few more details about data management needed e.g. which study personnel/roles will have access to the study data and for how long? How is the data stored and protected e.g. two factor authentication is required for access? Will the data be stored on servers located in Australia or the cloud? Can you comment on whether you may be underpowered for symptomatic infection outcomes, given these are less frequent (secondary outcomes) as part of the limitations please? It wasn't clear if the incidence rates you used for the power calculation were based on laboratory testing results including asymptomatic infection or symptomatic infection.
--	---

VERSION 1 – AUTHOR RESPONSE

Reviewer: 1

The importance of the study is clear, and the protocol is appropriate. I have three comments:

1. *Regarding inclusion criterium 4, committed not to take doxycycline as prophylaxis for the duration of the trial - this is a problem with regard to the 2023 Consensus Statement on doxycycline prophylaxis (Doxy-PEP) for the prevention of syphilis, chlamydia and gonorrhoea among gay, bisexual, and other men who have sex with men in Australia. Many participants may be eligible for doxycycline prophylaxis.*

Our response:

Regarding our inclusion criterium 4, "Committed not to take doxycycline as prophylaxis for the duration of the trial" we note that our trial started well before the release of the 2023 consensus statement. However, in Apr 2023 after initial results were publicised regarding Doxy-PEP and discussions with our protocol steering committee and ethics committee we notified site clinicians that they could prescribe doxycycline to study participants if they felt this was appropriate for their patients. We also requested that at each study visit that the site record the number of days participants have taken doxycycline since their last visit. We believe that this is in accordance with the 2023 Consensus Statement on doxycycline prophylaxis (Doxy-PEP).

The following text in has been added as a footnote to Box 1 at point 4 - Committed not to take doxycycline as prophylaxis for the duration of the trial*

*** In accordance with the Australasian Society for HIV, Viral Hepatitis and Sexual Health Medicine (ASHM) 2023 Consensus Statement on doxycycline prophylaxis (Doxy-PEP) (<https://ashm.org.au/about/news/doxy-pep-statement/>), site clinicians were advised that they could prescribe doxycycline to study participants if they felt this was appropriate for their patients, and the number of days participants have taken doxycycline is recorded at each visit.**

2. *Vaccine effectivity can be calculated from this trial. However vaccine effectivity is expected to be only 30-50%. To study the effectivity of the vaccine for public heath purposes it is important to know who is willing to take the vaccine. In the present study, no attempt is made to identify specific details of the population taking part in the study in comparison to those who were asked to take part but declined to participate. It could be useful to compare those groups by handing over a study questionnaire to persons eligible for the trial, but not inclined to take part.*

Our response:

We accept that such a questionnaire of people who declined to take part in the study might have proved interesting and will be particularly useful for public health purposes once vaccine effectiveness is better understood and the vaccine is offered outside of a trial setting. However, recruitment to the trial has closed, and we are not in a position to implement this at this stage. We do, however, collect data in the study electronic data capture system on the reason participants failed screening or were ineligible to be randomised or take part in the study. The reasons could be 'participant refused', 'failed inclusion or exclusion criteria', 'person will organise their own Meningococcal B vaccines outside of trial', and 'Other, specify'. These data will be summarised and may provide some insight.

We have added the following text in bold to describe this to the footnote of Table 2 which now reads; "a All screening assessments are to be completed within 14 days of baseline (the visit when the first dose of 4CMenB or matched placebo will be administered). **The reason participants failed screening or were ineligible to be randomised or take part in the study is recorded.**"

3. *NAAT specimens could be stored to sequence certain genes of interest of strains that could not be cultured. I suppose this will be done by the consortium but this could be stated if so.*

Our response:

We considered sequencing from NAAT samples, however this is not feasible due to logistical difficulties of acquiring the NAAT samples.

Reviewer: 2

This is an important well-designed study.

1. *Can you add more detail about data on participants who get a confirmed infection within the 4-week post dose 2 period please? Are they excluded or included?*

Our response:

Confirmed infections within the 4-week post dose 2 period will be excluded from the primary endpoint. However, we do collect data on these infections, rates will be summarised, and they will be included in secondary endpoints.

To clarify this point, we have added the text in bold and line 267 now reads "The primary efficacy endpoint is the incidence of first episode of *N. gonorrhoeae* infection detected by a NAAT, post month 4. A secondary endpoint will include the incidence of all episodes of *N. gonorrhoeae* infection detected by a NAAT, **including those within 4 weeks post vaccine dose 2.**"

2. *Vaccine efficacy equation should the percent sign be there? I don't think it should - the multiplication should simply be by 100 i.e. $100(1-(R_v/R_p))$ you could then note that VE is expressed as a percent. It's the same as adding any other unit e.g. ml to an equation.*

Our response:

We have modified the text at Line 283, which now reads as follows: "Vaccine efficacy (VE) is expressed as a percentage and will be defined as: $VE = 100\{1 - (r_v/r_p)\}$ "

3. *Maybe a few more details about data management needed e.g. which study personnel/roles will have access to the study data and for how long? How is the data stored and protected e.g. two factor authentication is required for access? Will the data be stored on servers located in Australia or the cloud?*

Our response:

The Data Management section in the manuscript has been modified to include additional details shown in bold below, and the section starting Line 231 now reads.

"Source documents include, but are not limited to participant medical records, laboratory reports, participant progress notes, pharmacy records and any other reports or records of procedures

performed in accordance with the protocol. Study data will be collected using an Electronic Data Capture (EDC) system, **Medrio. Medrio is a cloud-based**, web-enabled password-protected platform. **Data are stored using Google Cloud Platform facilities globally. Medrio is adherent to ICH GCP, FDA 21 CFR Part 11; EudraLex, Annex 11; GDPR; and HIPAA. All site data entry personnel will be required to pass training and assessments to be able to enter data and all will have their own confidential login credentials. The login password will be changed every 3 months.** Following each participant visit the designated site staff will complete the visit specific electronic case report form (eCRF) as soon as possible after the completion of the visit. **There will be no personal information or full identifiable information of any participants entered or stored in the study database.** Each participant will be assigned a unique Participant Identification Number (PIN). The PIN will be documented in the participant’s medical record and all study documents including the dispensing records. Participants also provide consent for the Project Team to acquire any other information and results from other health services for the purpose of the research via linkage with the Australian Collaboration for Coordinated Enhanced Sentinel Surveillance (ACCESS) system.³² Data from ACCESS will enable long-term follow-up after the completion of the study even if a participant moves to another clinic. **At The Kirby Institute, UNSW Sydney (Sponsor), designated research personnel have viewing access to the EDC to monitor data and conduct source data verification. The study data will be extracted after the database is closed at the end of the study. The study data will be kept for 15 years after study completion.**

4. *Can you comment on whether you may be underpowered for symptomatic infection outcomes, given these are less frequent (secondary outcomes) as part of the limitations please? It wasn't clear if the incidence rates you used for the power calculation were based on laboratory testing results including asymptomatic infection or symptomatic infection.*

Our response:

The primary endpoint includes both symptomatic and asymptomatic infections. We likely will be underpowered for symptomatic infections, though it is difficult to estimate power without knowing what proportion of infections will be symptomatic. We still believe it is worth analysing asymptomatic infections as a secondary endpoint, accepting the reduced power, and will interpret results accordingly.

We have added the following text in bold at line 259 to clarify this, “and a conservative *N. gonorrhoeae* incidence of 25 per 100 PY (based on recent Australian data for **symptomatic and asymptomatic infections in** GBM on PrEP (39 per 100 PY)⁷ and in HIV-positive GBM (29 per 100 PY)⁸”

VERSION 2 – REVIEW

REVIEWER	Paynter, Janine University of Auckland, School of Population Health
REVIEW RETURNED	01-Mar-2024
GENERAL COMMENTS	The authors have answered the previous review questions very well. I have no further comments.